# Unprecedented lattice volume expansion on doping stereochemically active $Pb^{2+}$ into uniaxially strained structure of $CaBa_{1-x}Pb_xZn_2Ga_2O_7$

Pengfei Jiang [1], Joerg C. Neuefeind [2], Maxim Avdeev[3,4], Qingzhen Huang[5], Mufei Yue[1], Xiaoyan Yang[6], Rihong Cong [1] & Tao Yang [1]✉

Lone pair cations like $Pb^{2+}$ are extensively utilized to modify and tune physical properties, such as nonlinear optical property and ferroelectricity, of some specific structures owing to their preference to adopt a local distorted coordination environment. Here we report that the incorporation of $Pb^{2+}$ into the polar "114"-type structure of $CaBaZn_2Ga_2O_7$ leads to an unexpected cell volume expansion of $CaBa_{1-x}Pb_xZn_2Ga_2O_7$ ($0 \leq x \leq 1$), which is a unique structural phenomenon in solid state chemistry. Structure refinements against neutron diffraction and total scattering data and theoretical calculations demonstrate that the unusual evolution of the unit cell for $CaBa_{1-x}Pb_xZn_2Ga_2O_7$ is due to the combination of the high stereochemical activity of $Pb^{2+}$ with the extremely strained $[Zn_2Ga_2O_7]^{4-}$ framework along the c-axis. The unprecedented cell volume expansion of the $CaBa_{1-x}Pb_xZn_2Ga_2O_7$ solid solution in fact is a macroscopic performance of the release of uniaxial strain along c-axis when $Ba^{2+}$ is replaced with smaller $Pb^{2+}$.

[1] College of Chemistry and Chemical Engineering, Chongqing University, Chongqing 401331, P. R. China. [2] Chemical and Engineering Materials Division, Spallation Neutron Source, Oak Ridge National Laboratory, Oak Ridge, TN 37831, USA. [3] Australian Nuclear Science and Technology Organization, Lucas Heights, NSW 2234, Australia. [4] School of Chemistry, The University of Sydney, Sydney, NSW 2006, Australia. [5] NIST Center for Neutron Research, National Institute of Standards and Technology, Gaithersburg, MD 20899, USA. [6] College of Materials Science and Engineering, Guilin University of Technology, Guilin, Guangxi 541004, P. R. China. ✉email: taoyang@cqu.edu.cn

 

one pair (LP) cations (Tl+, Pb2+, Bi3+, Te4+, Sb3+, I5+, etc.) with $(n-1)d^{10}ns^2$ electronic configurations are prone to adopt an asymmetric coordination environment and thus form a locally distorted structural unit possessing a dipolar moment. A long-range ordered alignment of such asymmetric units may break the inversion symmetry of the structure, possibly leading to a polar structure with intriguing properties. The strategy of using asymmetric units with LP cations is extensively utilized to design and synthesize noncentrosymmetric structures with enhanced nonlinear optical properties[1–7]. Moreover, LP cations in polar structures may lead to spontaneous electric polarizations and form ferroelectric structure, e.g., $BiFeO_3$[8]. In some cases, such LP cations-induced ferroelectricity can also trigger large negative thermal expansion (NTE) upon warming, which has also been extensively investigated in $PbTiO_3$-based perovskites[9–13].

Recently, the stereochemical activity owing to the LP electrons of $Sn^{2+}$ in $CsSnBr_3$[14], and $Pb^{2+}$ and $Sn^{2+}$ in rock-salt chalcogenides $PbS$[15], $PbTe$[15], and $SnTe$[16], is shown to cause a local symmetry lowering in a limited temperature range upon warming. This phenomenon is unique because it is an opposite behavior to that of most crystals showing symmetry lowering on cooling. X-ray and neutron total scattering experiments revealed that the progressively local distortion state in these compounds stems from the lattice expansion-induced dynamic off-center displacement of LP cations upon warming[14,15].

As described above, the incorporation of the stereochemically active LP cations into some structures brings not only intriguing physical properties but also some uncommon phenomena. In this work, we report another unprecedented phenomenon based on stereochemical activity of LP cations, i.e., cell volume expansion in solid solutions of $CaBa_{1-x}Pb_xZn_2Ga_2O_7$ $(0 \leq x \leq 1)$ when substituting $Ba^{2+}$ ($r = 1.61$ Å in 12-fold coordination) with smaller $Pb^{2+}$ ($r = 1.49$ Å in 12-fold coordination)[17]. Ostensibly, the abnormal cell volume expansion in $CaBa_{1-x}Pb_xZn_2Ga_2O_7$ is ascribed to the fact that the expansion of the $c$ axis (0.11 Å) is much larger than the contraction of the $a$ axis length (0.011 Å) in the space group $P6_3mc$. Neutron pair distribution function (nPDF) data analyses confirm that $CaBa_{1-x}Pb_xZn_2Ga_2O_7$ locally adopt a distorted orthorhombic structure ($Pna2_1$); however, this local distortion is not responsible for the abnormal cell volume expansion, as suggested by the Rietveld refinements based on high-resolution X-ray and neutron diffraction (ND) data. Density functional theory (DFT) calculations reveal that $Pb^{2+}$ $6s6p$ orbitals are highly hybridized with $O^{2-}$ $2p$ orbitals, which leads to the formation of a strong covalent bond and the resulting structural strain of the original $[Zn_2Ga_2O_7]^{4-}$ framework is released by significantly elongating the $c$ axis length.

## Results

**Unexpected cell volume expansion for $CaBa_{1-x}Pb_xZn_2Ga_2O_7$.** $CaBa_{1-x}Pb_xZn_2Ga_2O_7$ crystallize in the so-called "114"-type oxide structure, where the $[BaO_3]$ and $[O_4]$ layers form a mixed cubic and hexagonal closed-packing ionic structure with the stacking sequence of -CBABC-, leaving the octahedral and tetrahedral cavities partially occupied by $Ca^{2+}$ and $Zn^{2+}/Ga^{3+}$ cations, respectively[18]. This structure is more commonly viewed as a layered structure with alternating stacking of triangular and Kagome layers along the polar axis. Up to now, nitrogen ions and various metal cations could be incorporated into this structure[19–23]. This work is the report of $Pb^{2+}$ doping.

Powder X-ray diffraction (XRD) patterns for $CaBa_{1-x}Pb_xZn_2Ga_2O_7$ (Supplementary Fig. 1) indicate the phase purity as well as the high crystallinity of the samples. The anisotropic change of the lattice parameters can be readily visualized from

XRD, as the position for the (004) reflection evolves to lower angles, whereas the position for the (200) reflection shifts to higher angles on the $Pb^{2+}$-to-$Ba^{2+}$ substitutions. It is firmly corroborated by plotting the lattice parameters against the $Pb^{2+}$ content (Fig. 1), where the unit cell lattice contracts and expands along the $a$ and $c$-axes, respectively. Such an anisotropic change leads to an overall expansion of the unit cell volume for $CaBa_{1-x}Pb_xZn_2Ga_2O_7$ up to $x = 0.7$ and saturates thereafter. As discussed above, given the smaller ionic radius of $Pb^{2+}$ compared with that of $Ba^{2+}$, this is an unexpected and unique structural phenomenon in solid state chemistry. One might argue that the simple comparison of the cationic radii for $Ba^{2+}$ and $Pb^{2+}$ is not sufficient because the ionic size for LP cation-like $Pb^{2+}$ is not well defined owing to their typical distorted coordination environment. Therefore, a series of $Ba^{2+}$- and $Pb^{2+}$-containing compounds with identical structure types were compared in terms of their cell volumes per formula, as summarized in Supplementary Table 1. It is obvious that all the $Pb^{2+}$-containing compounds possess smaller volumes in comparison with those of $Ba^{2+}$-containing compounds, which confirms that the observed cell volume expansion in $CaBa_{1-x}Pb_xZn_2Ga_2O_7$ is unprecedented.

Furthermore, we synthesized $CaBa_{0.5}Sr_{0.5}Zn_2Ga_2O_7$ for comparison, because $Sr^{2+}$ ($r = 1.44$ Å) has a comparable cationic radius with $Pb^{2+}$[17], however, the $Sr^{2+}$-to-$Ba^{2+}$ substitution led to a significant cell volume contraction, i.e., V = 351.55 Å$^3$ for $CaBa_{0.5}Sr_{0.5}Zn_2Ga_2O_7$ and 356.73 Å$^3$ for $CaBaZn_2Ga_2O_7$. Accordingly, we conclude that it is not the cationic size that

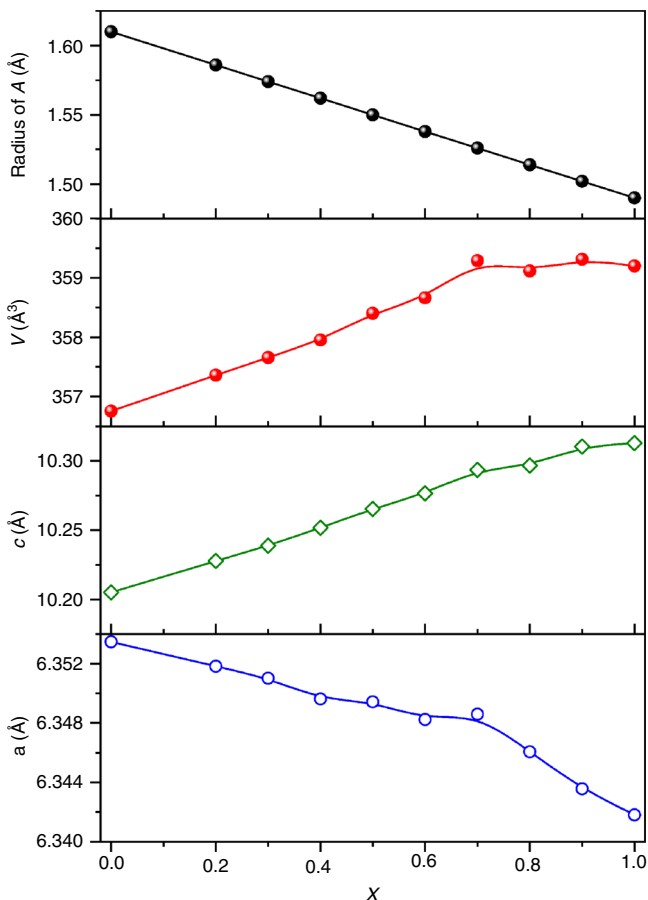

**Fig. 1 Lattice parameters.** Plots of the average A-site size and refined lattice parameters for $CaBa_{1-x}Pb_xZn_2Ga_2O_7$ along with the increasing Pb content. The average radius of A-site cation is calculated according to the equation: $r(A^{2+}) = (1-x) \, r(Ba^{2+}) + x \, r(Pb^{2+})$. Source data are provided as a Source Data file.

governs the lattice expansion in $CaBa_{1-x}Pb_xZn_2Ga_2O_7$, instead, it strongly points to the special characteristics of $Pb^{2+}$ ($6s^2$ LP electrons).

**Reciprocal space structure refinements.** Crystal structures for all solid solutions were investigated by Rietveld refinements using XRD data so as to reveal the origin of this unexpected cell volume expansion. $CaBaZn_2Ga_2O_7$ was used as the starting structural model ($P6_3mc$) for structure refinements. $Zn^{2+}$ and $Ga^{3+}$ could not be distinguished by X-ray scattering owing to their similar atomic form factors, thus $Zn^{2+}$ and $Ga^{3+}$ were treated as the same cation during the Rietveld refinements on XRD. As all the samples were found phase pure and metal disorder between (Ba, Pb) and (Ga, Zn) sites is implausible due to very large difference in size (>220%), the occupancy factors for $Ba^{2+}$ and $Pb^{2+}$ were fixed to the nominal values. Finally, a model with anisotropic atomic displacements (ADPs) was used to account for the local displacement disorder $Ba^{2+}/Pb^{2+}$ cations in $CaBa_{1-x}Pb_xZn_2Ga_2O_7$ ($0 < x < 1$) (see more detail in Supplementary Note 2). The ADPs for the (Ba/Pb) site were found elongated along the $c$ axis (Supplementary Table 2), which is consistent with the LP effect of $Pb^{2+}$. Moreover, the $U_{33}/U_{11}$ ratio evolution with $x$ proved a good indicator of the structural strain being relieved with increasing content of $Pb^{2+}$. At low doping levels, where the geometry of the crystal structure is determined mostly by $Ba^{2+}$, the local displacements mimicked by the ADPs are the largest (Supplementary Fig. 2). Further increasing content of $Pb^{2+}$ relieves the strain by stretching the $c$ axis and the ADPs become progressively isotropic. In the composition with the highest $x$, it is $Pb^{2+}$ that controls the crystal structure geometry and $c$ axis becomes too long for $Ba^{2+}$ and ADP becomes a flattened ellipsoid within the $ab$-plane, so $CaPbZn_2Ga_2O_7$ was described with a site splitting model. The refinements converged rapidly to give stable structures with reasonable crystallographic parameters and advantageous agreement factors for the solid solutions. The final Rietveld refinement patterns for solid solutions are presented in Supplementary Fig. 3. The resultant crystallographic data are summarized in Supplementary Tables 2 and 3.

To determine the crystal structure accurately, high-resolution constant wavelength ND data and Cu Kα1 data for representative compositions $CaBa_{1-x}Pb_xZn_2Ga_2O_7$ ($x = 0$, 0.5, and 1) were collected. By ND, the occupancy factors of $Ga^{3+}$ and $Zn^{2+}$ at T sites could be determined, because there is a large contrast in neutron scattering length between $Zn^{2+}$ (5.68 fm) and $Ga^{3+}$ (7.29 fm). Combined Rietveld refinements against both the ND and XRD data revealed that the occupancy factor for $Zn^{2+}$ at T1 site converged to ~ 0.21 for all compositions, which confirms that the evolution of cell parameters and cell volume as a function of $x$ is driven by $Pb^{2+}$ content, and not by Zn/Ga distribution, which remains constant across the series.

It is noteworthy that the local displacement of $Pb^{2+}$ does not cause any average structure symmetry lowering and the $CaBa_{1-x}Pb_xZn_2Ga_2O_7$ samples can be well described using $P6_3mc$. All these observations in $CaBa_{1-x}Pb_xZn_2Ga_2O_7$ are thus different from other isostructural oxides without stereochemically active cations, e.g., $MAZn_2Ga_2O_7$ ($M = Ca^{2+}$, $Sr^{2+}$; $A = Ba^{2+}$, $Sr^{2+}$), where successive symmetry lowering and cation disorder-order transitions were observed[18]. Such symmetry and cation disorder-order transitions might be observed for $CaBa_{1-x}Pb_xZn_2Ga_2O_7$ at lower temperatures. The plots of combined Rietveld refinements are presented in Fig. 2 and Supplementary Fig. 4. The final crystallographic data and selected interatomic distances are summarized in Supplementary Tables 4 and 5.

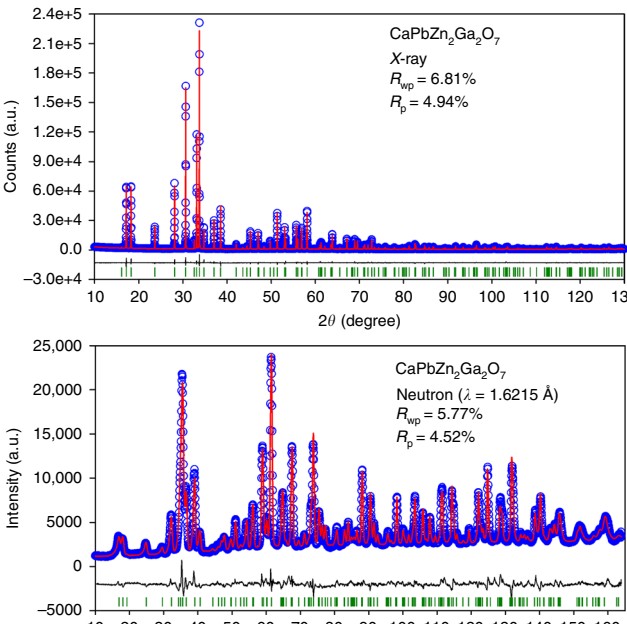

**Fig. 2 Reciprocal space structure refinements.** Combined Rietveld refinement plots of Cu Kα₁ XRD and constant wavelength ND data for $CaPbZn_2Ga_2O_7$ refined with the $P6_3mc$ model. The overall reliability factors for combined Rietveld refinements are $R_{wp} = 6.44\%$, $R_p = 4.77\%$. Source data are provided as a Source Data file.

**Real space structure refinements.** Combined Rietveld refinements using both XRD and ND data are powerful to analyze the average structure. On the other hand, the total scattering technology allows both the coherent and diffuse components of the XRD/ND pattern to be properly accounted for when modeling a crystal structure. We utilized nPDF analyses to reveal the evolution of local structural distortions and cationic ordering, which helps understand the unique cell volume expansion phenomenon in $CaBa_{1-x}Pb_xZn_2Ga_2O_7$.

The normalized structure functions and nPDFs for $CaBa_{1-x}Pb_xZn_2Ga_2O_7$ ($x = 0$, 0.5, and 1) are presented in Supplementary Fig. 5. Apparent shift of the positions of Pb/Ba−O and Zn/Ga−Zn/Ga pairs are observed for the nPDFs (Supplementary Fig. 5b), which is in line with the results deduced from XRD and ND. The peak for nearest Zn/Ga−O pairs is almost symmetric although the cationic size difference for $Zn^{2+}$ and $Ga^{3+}$ is large. Moreover, owing to the overlap of the positions between Zn/Ga−Zn/Ga and Pb/Ba−O pairs, the local Zn/Ga ordering cannot be visually observed. Apart from the change of peak positions, another noteworthy feature is the obvious decrease or increase in the magnitude of the atomic pairs with increasing $Pb^{2+}$ content, i.e., Zn/Ga−O and Ca−O pairs, indicating a wide range of atomic distributions. This observation suggests the $Pb^{2+}$ doping may lead to a local structure distortion.

Then real space refinements were performed against the neutron PDF data with the average structure model ($P6_3mc$) obtained from the combined refinement. However, this model does not reproduce well the peak at ~ 2.3 Å, which is corresponding to the Ca−O pairs (Fig. 3 and Supplementary Fig. 6). Moreover, this structure model also overestimates the Zn/Ga−O distances (at ~ 4.8 and 6.3 Å) by ~ 0.08 Å. These observations suggest that neither the $CaO_6$ nor $(Zn/Ga)O_4$ polyhedra extracted from the neutron PDF data can be well described by the average structure model, especially in the low $r$ range. This is a strong indication of the local structural

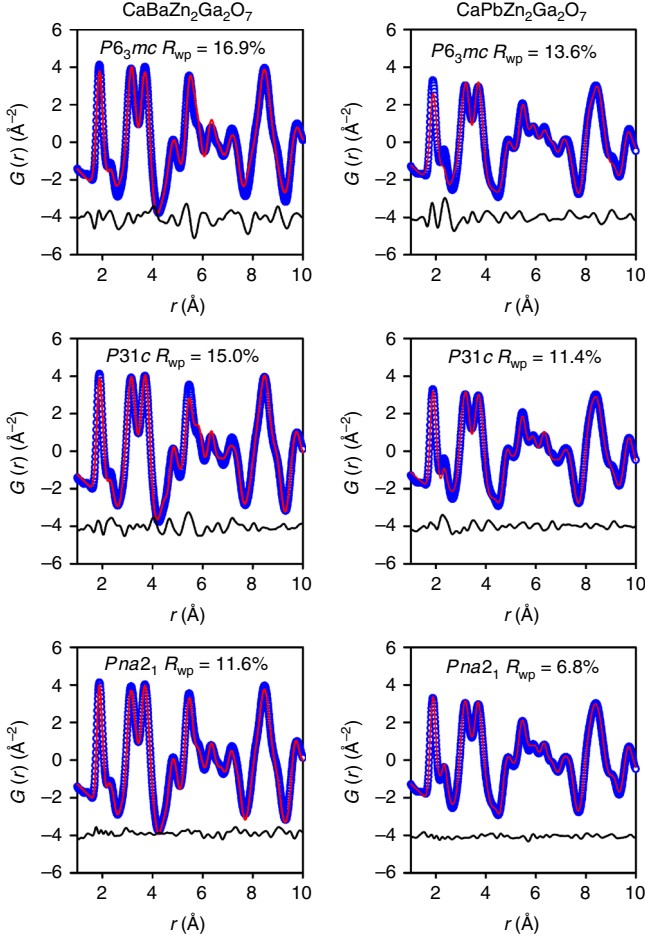

**Fig. 3 Real space structure refinements.** Real space Rietveld refinements of neutron PDF data for $CaBaZn_2Ga_2O_7$ and $CaPbZn_2Ga_2O_7$ with $P6_3mc$, $P31c$, and $Pna2_1$ models. Source data are provided as a Source Data file.

distortion in $CaBa_{1-x}Pb_xZn_2Ga_2O_7$. To gain an accurate picture of the local structure, structure models in the space group $P31c$ and $Pna2_1$, which allow the free distortion of $(Zn/Ga)O_4$ tetrahedra, were employed. As shown in Fig. 3 and Supplementary Fig. 6, the refinement using the $P31c$ model gives agreement similar to that of the $P6_3mc$ model, whereas the refinements using the $Pna2_1$ model provides an excellent fit for both peaks of Ca—O and Zn/Ga—O pairs. Therefore, the local structure for $CaBa_{1-x}Pb_xZn_2Ga_2O_7$ exhibits a lower symmetry, $Pna2_1$. Because both $Pb^{2+}$-containing and -free compounds are locally distorted, such a local structure distortion unambiguously ascribes to $Zn^{2+}/Ga^{3+}$ disordering, rather than the $Pb^{2+}$-to-$Ba^{2+}$ substitutions. It was recently reported that a similar cation inversion disordering induced a local structure distortion in spinel $Mg_{1-x}Ni_xAl_2O_4$ and $CuMn_2O_4$[24,25], which were also probed by the neutron total scattering technique. Here, the refined nanoscale structures for $CaBa_{1-x}Pb_xZn_2Ga_2O_7$ ($x = 0$ and 1) (Supplementary Fig. 7) demonstrate that the local orthorhombic distortion is not significant, further indicating the distortion on the long range would be also insignificant. The lattice parameters extracted from neutron PDF analyses also exhibit an anisotropic shrinkage and expansion within the $ab$-plane and along the $c$ axis (Supplementary Fig. 8), respectively, which results in the increase of the unit cell volume with the increasing $Pb^{2+}$ content.

Combined reciprocal and real space Rietveld refinements against ND and PDF data were further performed to reveal the

long-range structure symmetry. In the long range, only the $P6_3mc$ and $P31c$ models were considered because the reflection conditions for the $Pna2_1$ model are not consistent with the ND data. As shown Supplementary Figs. 9–11, all the PDF peaks for $CaBa_{1-x}Pb_xZn_2Ga_2O_7$ ($x = 0$, 0.5, 1) in long range (30–50 Å) can be well produced by the $P6_3mc$ model with reliable factors comparable with or even better than that of the low symmetry $P31c$ model, demonstrating the long-range structure symmetry for $CaBa_{1-x}Pb_xZn_2Ga_2O_7$ is $P6_3mc$, which is in good agreement with the combined Rietveld analysis of XRD and ND data.

**Structure evolution**. Figure 4 and Supplementary Fig. 12 give the comparison crystal structures for $CaBa_{1-x}Pb_xZn_2Ga_2O_7$ ($x = 0$ and 1) and $CaBa_{0.5}Sr_{0.5}Zn_2Ga_2O_7$. Apparently, $Ba^{2+}/Sr^{2+}$ in $CaBaZn_2$-$Ga_2O_7$ and $CaBa_{0.5}Sr_{0.5}Zn_2Ga_2O_7$ are located in the same $ab$-plane defined by O3 anions. In contrast, the $Pb^{2+}$ cations in $CaPbZn_2$-$Ga_2O_7$ show stereochemical activity with a significant displacement along the polar $c$ axis, which can be also deduced from the change of $Ba^{2+}/Pb^{2+}$−O interatomic distances in $CaBa_{1-x}Pb_xZn_2Ga_2O_7$ ($0 \leq x \leq 1$). As shown in Fig. 5a, the difference between the group of $Ba^{2+}/Pb^{2+}$−O1 bond lengths becomes larger when increasing the $Pb^{2+}$-content. The $Ba^{2+}/Pb^{2+}$−O3 bond also exhibits an increasing trend (Fig. 5b). Such evolution of $Ba^{2+}/Pb^{2+}$−O bond lengths indicates that the $Pb^{2+}$ cation displaces from the center of the $[Ba/PbO_{12}]$ dodecahedron rather than merely attracts the apical oxygen atom closer as in the case of $CaBa_{0.5}Sr_{0.5}Zn_2Ga_2O_7$ (Supplementary Table 3).

The calculated average deviation distance ($D$) for $Ba^{2+}/Pb^{2+}$ continuously increases along with the increase of the $Pb^{2+}$ content (Fig. 5c). The largest $D$ value ~ 0.5 Å is observed in $CaPbZn_2Ga_2O_7$. Such a significant deviation of $Pb^{2+}$ from the O3-plane towards to O1 results in a very irregular coordination environment for $Pb^{2+}$. The shortest Pb—O bond length (2.337 Å) in $CaPbZn_2Ga_2O_7$ is comparable with the values in $Ca_2PbGa_8O_{15}$ (2.29 Å)[26], $PbTiO_3$ (2.51 Å)[27], and $PbRuO_3$ (2.50 Å)[28], where the Pb $6s6p$ orbitals are all strongly hybridized with O $2p$ orbitals. So it is expected that the $Pb^{2+}$ $6s6p$ orbitals in $CaPbZn_2Ga_2O_7$ are also strongly hybridized with O $2p$ orbitals, which is further corroborated by DFT calculations. For example, the density of states analyses for $CaPbZn_2Ga_2O_7$ (Supplementary Fig. 13) indicate that both the bottom of the conduction band and the top of the valence band comprise the $Pb^{2+}$ $6s6p$ states, which is a solid evidence of the hybridization between Pb $6s6p$ and O $2p$ orbitals.

## Discussion

As discussed above, both real space and reciprocal data analysis decipher that the unprecedented cell volume expansion for $CaBa_{1-x}Pb_xZn_2Ga_2O_7$ ($0 \leq x \leq 1$) from nanoscale to a long-range scale is ascribed neither to the local structure distortion nor the cationic ordering. All these results deduced from the structure refinements point to the stereochemical activity of $Pb^{2+}$. LP induced displacement along a polar axis is commonly observed for many $Bi^{3+}$ and $Pb^{2+}$-containing perovskites, i.e., $BiMO_3$[29–31] and $PbMO_3$ ($M$ = transition metals)[32,33], which always results in a large tetragonality with $c/a > 1.0$. This enhanced tetragonality is closely related to the highly hybridization between $Pb^{2+}$ $6s^2$ and $O^{2-}$ $2p$ orbitals, which is indeed observed in $(x)BiFeO_3$-$(1-x)$ $PbTiO_3$[34] and $Pb_{0.8-x}La_xBi_{0.2}VO_3$[12] solid solutions. Here in $CaBa_{1-x}Pb_xZn_2Ga_2O_7$, the enhancement of $c/a$ values is also accompanied with the increase of the stereochemical activity of $Ba/Pb^{2+}$ cations (or enhancement of $Ba^{2+}/Pb^{2+}$−O covalency) (Fig. 6a). This observation corroborates that the structural anisotropy as well as the cell volume expansion for $CaBa_{1-x}Pb_xZn_2Ga_2O_7$ is dominated by the stereochemically

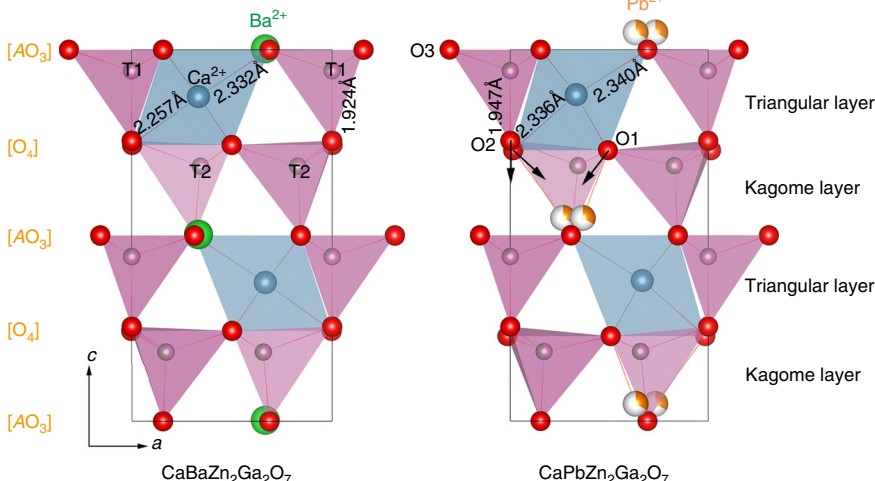

**Fig. 4 Crystal structures for CaBaZn₂Ga2O₇ and CaPbZn₂Ga₂O₇.** Crystal structures were obtained from the combined Rietveld refinements on ND and XRD data with the $P6_3mc$ model. In both structures T1 and T2 sites are co-occupied by $Ga^{3+}$ and $Zn^{2+}$ but with different occupancies, where T1 site is dominated by $Ga^{3+}$ with an occupancy factor of 0.79(3), accordingly T2 site is mainly occupied by $Zn^{2+}$ with an occupancy factor of 0.60(2). The black arrows represent the shift direction of oxygen atoms.

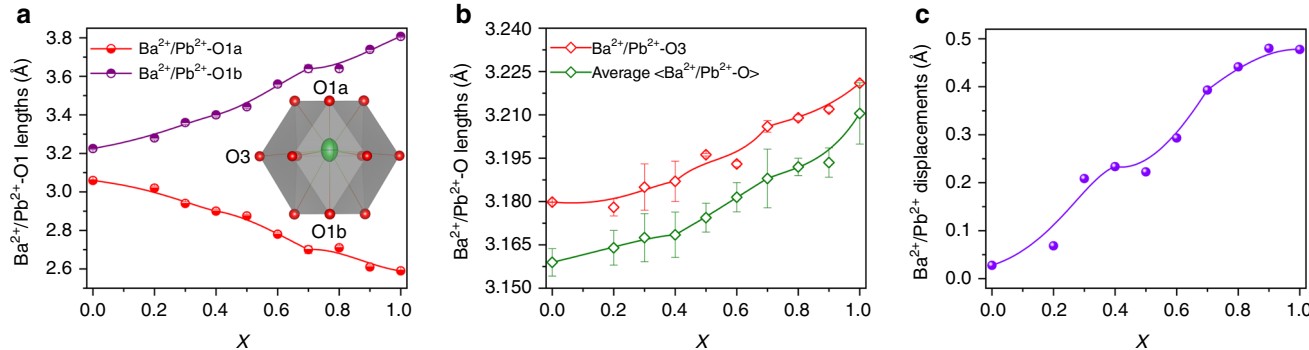

**Fig. 5 Evolution of Ba/Pb−O bond lengths and Ba/Pb displacements.** Plots of $Pb^{2+}/Ba^{2+}−O1$ bond lengths **a**, $Pb^{2+}/Ba^{2+}−O3$ and average $< Pb^{2+}/Ba^{2+}−O >$ bond lengths **b**, and average $Pb^{2+}/Ba^{2+}$ displacement distance **c** against the Pb content in $CaBa_{1−x}Pb_xZn_2Ga_2O_7$. The inset shows the coordination environment of $Ba^{2+}/Pb^{2+}$ in $CaBa_{1−x}Pb_xZn_2Ga_2O_7$ (0 < x < 1) obtained from Rietveld refinements against XRD data with the $P6_3mc$ model, where a highly anisotropic thermal motion along $c$ axis can be visually observed. For $x = 0$, 0.5, and 1, the parameters are obtained from combined Rietveld refinement against both XRD and ND data. Source data are provided as a Source Data file.

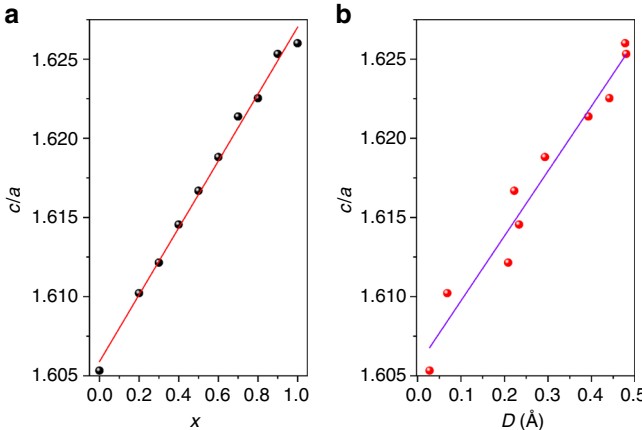

**Fig. 6 Evolution of c/a values.** Plots of $c/a$ values against **a** the doping Pb content and **b** the stereochemical lone pair activity (average displacements for $Ba^{2+}/Pb^{2+}$ cations) in $CaBa_{1−x}Pb_xZn_2Ga_2O_7$. For $x = 0$, 0.5, and 1, the parameters are obtained from combined Rietveld refinement against both XRD and ND data. Source data are provided as a Source Data file.

active LP electrons of $Pb^{2+}$. Moreover, the $c/a$ value for $CaPbZn_2Ga_2O_7$ can be reduced upon heating (Supplementary Fig. 14), which is similar to the $PbTiO_3$-based ferroelectricity-NTE materials[9,10]. This observation further reinforces the hypothesis that the stereochemical activity of $Pb^{2+}$ is responsible for the anisotropic lattice expansion. However, NTE is not realized for $CaPbZn_2Ga_2O_7$ and an expansion of the $c$ axis rather than contraction was observed, suggesting the unique structure for "114" oxides should be also responsible for the lattice volume expansion.

To further elucidate this conjecture, we also prepared the solid solutions of $Ba_{1−y}Pb_yTiO_3$ ($y = 0$, 0.5, and 1). $Pb^{2+}$-to-$Ba^{2+}$ substitutions in $Ba_{1−y}Pb_yTiO_3$ also lead to an anisotropic change of the cell dimensions as expected, however the cell volume shrinks (Supplementary Figs. 15 and 16). For $Ba_{1−y}Pb_yTiO_3$, the enhancement of A-site stereochemical activity also leads to an obvious off-center shift of $Ti^{4+}$ along the $c$ axis, which in turn results in a significant contraction of the $a$ axis, which compensates $c$ axis elongation, so that the overall cell volume decreases. In contrast, the crystal structure for "114" oxides is highly strained in comparison with the flexible framework of perovskite, which can be deduced by the $c/a$ value and bond valence sum

(BVS) calculations[35,36]. The $c/a$ value for $CaBaZn_2Ga_2O_7$ ($\sim 1.605$) is much smaller than the ideal closed-packing value ($c/a_{ideal} = 1.633$) and the BVS for $Ca^{2+}$ in $CaBaZn_2Ga_2O_7$ is estimated to be 2.48(1). All that suggests the fact that $[Zn_2Ga_2O_7]^{4-}$ anionic framework is highly strained along the $c$ axis (or $c$ axis is over-compressed), especially for the triangular layers because $Ca^{2+}$ is seriously over-bonded (Fig. 4).

Structure strain is usually released through structure symmetry lowering and polyhedral distortion/rotation, which was observed indeed in numerous "114" oxides such as $LnBaFe_4O_7$ ($Ln =$ Y, Dy–Lu)[23,37], $LnBaCo_4O_7$ ($Ln =$ Ca, Y, Tb–Lu)[38–40], and $MAZn_2Ga_2O_7$ ($M = Ca^{2+}$, $Sr^{2+}$; $A = Sr^{2+}$, $Ba^{2+}$)[18]. Herein, the structure strain of $CaBa_{1-x}Pb_xZn_2Ga_2O_7$ is released without a significant distortion of the $[Zn_2Ga_2O_7]^{4-}$ framework, that is, the stereochemically active LP effect of $Pb^{2+}$ helps the release of the structure strain. In detail, the LP active $Pb^{2+}$ displaces from the center of the $[Ba/PbO_{12}]$ dodecahedron to form strong covalency bonds with O1, which attenuates the covalency of $[Zn_2Ga_2O_7]^{4-}$ anionic framework and drives displacement of O1 and O2 downwards along the $c$ axis (Fig. 4b), resulting in a relief of uniaxial structure strain through a significant expansion of the $c$ axis. Such a significant expansion for $c$ axis further leads to an enhancement of the $c/a$ value to 1.626 for $CaPbZn_2Ga_2O_7$ and improvement of BVS value for $Ca^{2+}$ in $CaPbZn_2Ga_2O_7$ to 2.31 (4), suggesting the anisotropic lattice-change and cell volume expansion is cooperative with the release of uniaxial structure strain along the $c$ axis in $CaBa_{1-x}Pb_xZn_2Ga_2O_7$. Thus, the expansion of polar $c$ axis for $CaPbZn_2Ga_2O_7$ upon heating, which is different from the behavior of NTE materials, is also understandable. Finally, we can conclude that both the anisotropic chemical pressure induced by the highly stereochemical active LP electrons of $Pb^{2+}$ and the uniaxial structure strain along $c$ axis are responsible to the unprecedented cell volume expansion in $CaBa_{1-x}Pb_xZn_2Ga_2O_7$ solid solutions. In summary, substitution of $Ba^{2+}$ in $CaBaZn_2Ga_2O_7$ with the smaller but stereochemically active $Pb^{2+}$ results in an unprecedented lattice volume expansion, which has not been observed in solid state chemistry to the best of our knowledge. Both reciprocal and direct space XRD and ND data analysis were utilized to decipher the origin of this unique phenomenon on the scale of both local and average structure of $CaBa_{1-x}Pb_xZn_2Ga_2O_7$. The results revealed that the LP electrons of $Pb^{2+}$ are highly stereochemically active in this uniaxial greatly strained framework. Further DFT calculations revealed that the $Pb^{2+}$ $6s6p$ orbitals are highly hybridized with $O^{2-}$ $2p$ orbitals. The combined effect of these factors is the displacement of $Pb^{2+}$ along the $c$ axis, resulting in the release of structure strain associated with the axial cell expansion that is not compensated by contraction of the $ab$-plane and in turn produces the overall cell volume expansion of $CaBa_{1-x}Pb_xZn_2Ga_2O_7$. In general, our findings open new opportunities to use stereochemically active cations ($Pb^{2+}$, $Sn^{2+}$, $Bi^{3+}$, etc.) to tune crystal structural strain, which has important role in ferroic materials.

## Methods

**Sample preparation**. Polycrystalline samples of $CaBa_{1-x}Pb_xZn_2Ga_2O_7$ ($0 \le x \le 1$) and $CaSr_{0.5}Ba_{0.5}Zn_2Ga_2O_7$ were prepared by conventional high temperature solid state reactions. Calcium carbonate ($CaCO_3$, 99.99%), barium carbonate ($BaCO_3$, 99.99%), strontium carbonate ($SrCO_3$, 99.99%), lead oxide (PbO, 99.9%), zinc oxide (ZnO, 99.99%) and gallium oxide ($Ga_2O_3$, 99.99%) were used as starting materials. All the raw materials except for PbO were heated at 500 °C for 10 h before being weighted, in order to remove any adsorbed moisture. Stoichiometric raw materials were mixed and ground in an agate mortar and pre-heated at 800 °C for 10 h to decompose the carbonate. After this initial calcination, the resultant powder samples were re-ground thoroughly by hands and pressed into a pellet ($\varphi = 13$ mm). With increasing the doping content of $Pb^{2+}$, the synthetic temperatures of $CaBa_{1-x}Pb_xZn_2Ga_2O_7$ decrease owing to the relative low melting point of PbO. The pellets were heated in the range of 860–1100 °C for 60 h with intermediate re-grindings. Moreover, for the preparation of $CaBa_{1-x}Pb_xZn_2Ga_2O_7$ ($0.7 \le x \le 1$), additional 10 mg PbO should be added to compensation the volatilization of PbO after every cycle of calcination. $CaSr_{0.5}Ba_{0.5}Zn_2Ga_2O_7$ was prepared by heating raw materials at 1100 °C for 45 h with intermediate re-grindings.

**Structure characterizations**. The phase purity of the samples can be ensured by powder XRD. XRD was performed on a PANalytical Empyrean powder diffractometer equipped with a PXIcel 1D detector. Room temperature constant ND data for $CaBa_{1-x}Pb_xZn_2Ga_2O_7$ ($x = 0$ and 0.5) ($\lambda = 2.0775$ Å) and $CaPbZn_2Ga_2O_7$ ($\lambda = 1.6215$ Å) were collected at the BT-1 high-resolution ND diffractometer at the NIST Center for Neutron Research (NCNR) and ECHIDNA high-resolution powder diffractometer at the OPAL research facility (Lucas Heights, Australia)[41], respectively. Combined Rietveld refinements on ND and X-ray data were performed using the TOPAS-Academic V6 software[42].

Neutron total scattering experiments were performed at room temperature utilizing the nanoscale ordered materials diffractometer (NOMAD) at the spallation neutron source located at Oka Ridge National Laboratory. About 150 mg of each sample were loaded into a 2 mm diameter quartz capillary for measurements at room temperature with a collection time of $\sim 2$ h per sample. The PDF, G(r), was obtained through the Fourier transformation of S(Q) with Q value between 0.1 and 31.4 Å.

**DFT calculations**. Theoretical study of $CaPbZn_2Ga_2O_7$ was carried out using Vienna ab-initio simulation package (VASP)[43]. The projector augmented-wave method implemented in the VASP code was utilized to describe the interaction between the ionic cores and the valence electrons[44]. The generalized gradient approximation parameterized by Perdew, Burke, and Ernzerhof was employed to describe the exchange-correlation potential in the standard DFT calculations[45]. For single point energy and density of states, a cutoff energy of 500 eV for the plane-wave basis and $13 \times 13 \times 7$ Monkhorst-Pack G-centered $k$-point meshes were employed.

## Data availability

All relevant data that support the results of this study are available from the corresponding author upon request.

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

## Acknowledgements

This work is financially supported by the National Science Foundation of China (21805020, 21671028, 21771027), Natural Science Foundation of Chongqing (cstc2019jcyj-msxmX0330), Fundamental Research Funds for Central Universities (2019CDQYWL009, 2019CDXYHG0013), Chongqing Postdoctoral Science Special Foundation (XmT2018004), and Postdoctoral Research Foundation of China (2018M643402). A portion of this research used resources at the Spallation Neutron Source, a DOE Office of Science User Facility operated by the Oak Ridge National Laboratory. We also thank Professor Xiaojun Kuang in Guilin University of Technology for data collection. This manuscript has been authored by UT-Battelle, LLC, under contract DE-AC05-00OR22725 with the US Department of Energy (DOE). The US government retains and the publisher, by accepting the article for publication, acknowledges that the US government retains a nonexclusive, paid-up, irrevocable, worldwide license to publish or reproduce the published form of this manuscript, or allow others to do so, for US government purposes. DOE will provide public access to these results of federally sponsored research in accordance with the DOE Public Access Plan (http://energy.gov/downloads/doe-public-access-plan).

## Author contributions

T.Y. and P.J. conceived the ideal and design the project. P.J. synthesized the samples, analyzed the data and wrote the manuscript. J.C.N. collected the neutron total scattering data. M.A. and Q.H. collected the ND data. M.Y. performed the DFT calculations. X.Y. collected variable temperature XRD data. T.Y., R.C., and M.A. revised the manuscript. All authors discussed the results and commented on the paper.

## Competing interests

The authors declare no competing interests.
