## [Peer Review File · Nature Communications]

Reviewers' comments:

Reviewer #1 (Remarks to the Author):

Jiang and coworkers have submitted a work on the unconventional volume expansion observed for the complex oxide solid solution $\text{CaBa}_{1-x}\text{Pb}_x\text{Zn}_2\text{Ga}_2\text{O}_7$. The authors have used standard X-ray and neutron diffraction and Rietveld refinement combined with total neutron scattering / pdf method and DFT simulations. The combined methodology is not unique and is frequently used in state-of-the-art solid-state chemistry/materials chemistry, and at first glance this work may appear as "standard" crystallographic characterization of a solid solution series. However, as the title and the manuscript reflect the volume expansion observed by substituting the large Ba^{2+} cation with the smaller Pb^{2+} cation is rather unique example in solid state chemistry. The findings reported in the manuscript do therefore merit publication in Nature Communication. There are however some comments the authors are recommended to consider before publication.

Include the space group used in the various parts of the study, including figure captions and tables. This is important since different space groups are used throughout the study.

The crystal structure of the solid solutions was determined to be none-centrosymmetric by PDF and the materials should be piezoelectric/pyroelectric and possibly also ferroelectric at least at a local scale. Did the author look for possible ferroelectric/piezoelectric properties? Moreover, it would be of considerable value to expand the DFT calculations to determine the magnitude of polarization for selected compositions in the none-centrosymmetric space group by Berry phase calculations (or calculations by point charge model). Transitions from a non-polar to a polar state are typically accompanied with a volume expansion, and it is therefore of interest to investigate if one also observes an anomaly in polarization as observed in the molar volume versus composition.

The authors have provided important non-ambient X-ray diffraction data demonstrating that there is no dramatic contraction of the c-lattice parameter with increasing temperature, but there is a significant reduction of the thermal expansion along c at elevated temperatures. This effect is however not strong enough to result in negative expansion behaviour, found from many Pb-containing ferroelectrics. This suggest that the volume expansion due to Pb lone pairs are rather stable although a contraction of the c-axis can be inferred from the high temperature X-ray data. It would also be of interest to extend these data to above 800 C where one should see that the thermal expansion along c-axis "recover". The temperature dependence of the lattice parameters for $\text{CaBaPbZn}_2\text{Ga}_2\text{O}_7$ is quite important and deserve to be a part of the main report and not only in supporting information.

Reviewer #2 (Remarks to the Author):

This manuscript reports expansion of unit cell volume when Ba^{2+} ions are replaced by Pb^{2+} ions. The authors claim that the behavior of unit cell expansion is unprecedented based on ionic size consideration that the Pb^{2+} ions are smaller than Ba^{2+} ions. The local lattice distortion due to lone pair effect is well known in the literature. In the case of substitution of cations with lone pair, the ionic radii consideration does not work well as nature of distortion is not taken in to consideration. Therefore I do not see any surprise in the behavior of cell volume. I find that this manuscript does not contain any novelty and therefore do not find it suitable for publication in Nature Communications.

I have the following comments on the manuscript.

While there is a large difference in the scattering factor of Ba and Pb, why the occupancy factor of Ba and Pb were fixed? The authors suggest the disorder is responsible for high thermal parameters for Ba/Pb site. But I see that the U is high for $x=1$ as well. Is it because of large asymmetric distortion around Pb-ions?

Reviewer #3 (Remarks to the Author):

The manuscript Unprecedented Lattice Volume Expansion on Doping Stereochemically Active Pb²⁺ into Rigid Structure of CaBa_{1-x}Pb_xZn₂Ga₂O₇ reports extensive structural characterisation of the title solid solution using X-ray and neutron diffraction/PDF supported by DFT calculations. The cell volume expansion claimed by the authors is certainly a peculiar behaviour of this original solid solution. The evidences laid out to support this surprising feature are convincing and the role of the Pb²⁺ lone pair is clearly identified by the authors. I believe that these results are of interest for the modification of all structures build upon BaO₃ and O₄ closed packing and therefore of great interest to the solid state community. However, in the present form I cannot recommend this article for publication in nature communication without major revision.

The structure of the present paper is a bit confusing, the problem in my opinion is that the three structural studies are presented separately. This results in three different structural models to describe the same structure. In the first part of the manuscript all the composition are analyzed using XRD considering Ba/Pb and Ga/Zn mixed sites. Then in the second part of the manuscript, using neutron diffraction the authors are able to show that Ba and Pb are not on the same site (but keep correlated occupancies) and that the ratio of Zn/Ga is constant for the compositions $x = 0, 0.5, 1$. Finally, the orthorhombic distortion is used to explain the local structure by nPDF, but the authors do not mention which model fits best the long range data (according to the previous assumptions, the Pna21 model should be worse than P63mc). Therefore I suggest that all structural data be homogenized before publication to fit the most accurate model determined by XRD+NPD combined refinement or the nPDF eventually. Additionally, I would strongly recommend to used O₃ as the origin of the polar space group rather than Ba/Pb to match with the text, i.e. Pb pops out of the plan instead of the whole layer moves away from Pb.

The crystal chemistry of "114" oxides is not sufficiently described, the lower symmetry space groups (P31c and Pna21) used by the authors have all been described for the cobaltite series LnBaCo₄O₇ with Ln = Ca, Y, Tb – Lu. In addition, LnBaFe₄O₇ with Ln = Y, Dy – Lu, a cubic polymorph, based on a different stacking of the same elemental layer, has also been described to go from F4-3m to I-4 to P21. The numerous phase transitions observed in these systems, as well as the local distortion evidence by nPDF, contradict the claim by the author of a "rigid structure" in the ab plan.

Few additional question and remarks

Fig 1: How is the size of the A-site calculated? And why is it so linearly dependent on the composition despite the accidents in the evolution of the cell parameters?

Fig 2: The legends are not properly described. Both curves in Fig 2-a have the same legend, it might be necessary to differentiate the two O₁ position. I suppose that <Ba-O> is the average distance, but this is not clearly stated. Finally these data correspond to a structural model proved to be wrong by neutron powder diffraction, Ba and Pb do not occupy the same site...

Fig 6: what is "the stereochemical lone pair activity"? I cannot find any details on this parameter.

Fig S8: I am not sure how this figure was obtained but if they are mode decomposition then the possible movement in Pna21 are a combination of the P31c and Pna21 space group.

Victor DUFFORT

Reviewers' comments:

Reviewer #1 (Remarks to the Author):

Jiang and coworkers have submitted a work on the unconventional volume expansion observed for the complex oxide solid solution $\text{CaBa}_{1-x}\text{Pb}_x\text{Zn}_2\text{Ga}_2\text{O}_7$. The authors have used standard X-ray and neutron diffraction and Rietveld refinement combined with total neutron scattering/pdf method and DFT simulations. The combined methodology is not unique and is frequently used in state-of-the-art solid-state chemistry/materials chemistry, and at first glance this work may appear as “standard” crystallographic characterization of a solid solution series. However, as the title and the manuscript reflect the volume expansion observed by substituting the large Ba^{2+} cation with the smaller Pb^{2+} cation is rather unique example in solid state chemistry. The findings reported in the manuscript do therefore merit publication in Nature Communication. There are however some comments the authors are recommended to consider before publication.

Include the space group used in the various parts of the study, including figure captions and tables.

This is important since different space groups are used throughout the study.

Response: Thanks for your suggestion. The space group is now included in figures, captions or tables.

The crystal structure of the solid solutions was determined to be none-centrosymmetric by PDF and the materials should be piezoelectric/pyroelectric and possibly also ferroelectric at least at a local scale. Did the author look for possible ferroelectric/piezoelectric properties? Moreover, it would be of considerable value to expand the DFT calculations to determine the magnitude of polarization for selected compositions in the none-centrosymmetric space group by Berry phase calculations (or

calculations by point charge model). Transitions from a non-polar to a polar state are typically accompanied with a volume expansion, and it is therefore of interest to investigate if one also observes an anomaly in polarization as observed in the molar volume versus composition.

Response: This is quite a good comment. In fact, we did measure the piezoelectric and ferroelectric properties of $\text{CaPbZn}_2\text{Ga}_2\text{O}_7$, however no obvious response could be obtained which suggests that the local Pb^{2+} displacements are not coherent over the long-range and the structure is anti-ferroelectric, rather than ferroelectric on average. At the same time, the very high uniaxial strain discussed in detail in the text is very high and locks in the Pb^{2+} cations too strongly, so that they cannot be switched by external field up to the breakdown value.

The authors have provided important non-ambient X-ray diffraction data demonstrating that there is no dramatic contraction of the c-lattice parameter with increasing temperature, but there is a significant reduction of the thermal expansion along c at elevated temperatures. This effect is however not strong enough to result in negative expansion behavior, found for many Pb-containing ferroelectrics. This suggest that the volume expansion due to Pb lone pairs are rather stable although a contraction of the c-axis can be inferred from the high temperature X-ray data. It would also be of interest to extend these data to above 800 °C where one should see that the thermal expansion along c-axis “recover”. The temperature dependence of the lattice parameters for $\text{CaBaPbZn}_2\text{Ga}_2\text{O}_7$ is quite important and deserve to be a part of the main report and not only in supporting information.

Response: It is quite a good question. We re-measured the in-situ high temperature XRD data up to 1000 °C for $\text{CaPbZn}_2\text{Ga}_2\text{O}_7$, but this sample slowly decomposes into ZnO, ZnGa_2O_4 and other solid phases when above 900 °C. $\text{CaPbZn}_2\text{Ga}_2\text{O}_7$ almost completely decomposes at 975 °C (see Figure R1). As shown in Figure S14, the c-axis linearly expands from room temperature to 350 °C, then it

almost keeps a constant until the decomposition. No “recover” of the expansion of *c*-axis is observed.

As indicated in the main text, $\text{CaBa}_{1-x}\text{Pb}_x\text{Zn}_2\text{Ga}_2\text{O}_7$ is highly strained along the *c*-axis. At low temperature (≤ 350 °C), the expansion of the *c*-axis helps the release of the structure strain along the *c*-axis, which leads to a slight increase of *c/a* value. At elevated temperature (> 350 °C), the lone pair effect of Pb^{2+} is dominant and thus leads to a suppression of the expansion of the *c*-axis. In the meantime, the expansion of the *a*-axis become more obvious, resulting a significant decrease of *c/a*. These observations manifest that the lone pair effect of Pb^{2+} in $\text{CaPbZn}_2\text{Ga}_2\text{O}_7$ is indeed very stable, as suggested by Reviewer #1.

Figure R1. Selected in-situ high temperature XRD patterns for $\text{CaPbZn}_2\text{Ga}_2\text{O}_7$.

Reviewer #2 (Remarks to the Author):

This manuscript reports expansion of unit cell volume when Ba^{2+} ions are replaced by Pb^{2+} ions.

The authors claim that the behavior of unit cell expansion is unprecedented based on ionic size

consideration that the Pb^{2+} ions are smaller than Ba^{2+} ions. The local lattice distortion due to lone pair effect is well known in the literature. In the case of substitution of cations with lone pair, the ionic radii consideration does not work well as nature of distortion is not taken in to consideration. Therefore, I do not see any surprise in the behavior of cell volume. I find that this manuscript does not contain any novelty and therefore do not find it suitable for publication in Nature Communications.

Response: We agree with the reviewer that “the local lattice distortion due to lone pair effect is well known in the literature”. However, as we show in the revised manuscript, there is no a single example of the overall structure expansion on substitution of Ba^{2+} by Pb^{2+} . We added a list of isostructural compositions, which clearly shows that Pb^{2+} analogs are always smaller than Ba^{2+} counterparts.

In Table S1 (also see below) we provide the structure information and the cell volume (per formula) of a series of mineral structures that both accommodate Ba^{2+} and Pb^{2+} . Accordingly, all the Pb^{2+} -containing compounds exhibit smaller lattice volumes in comparison with that of Ba^{2+} -based compounds. So the observed lattice volume expansion in $\text{CaBa}_{1-x}\text{Pb}_x\text{Zn}_2\text{Ga}_2\text{O}_7$ solid solutions is out of expectation.

Table S1. The comparison of cell volumes for isostructural compounds containing Ba^{2+} and Pb^{2+} .

Compound	Structure type	Cell volume (\AA^3)	Volume per formula (\AA^3)
BaWO_4	Scheelite	361.85	90.25
PbWO_4	Scheelite	357.30	89.325
BaGa_2O_4	Tridymite	866.58 ($Z = 8$)	108.323
PbGa_2O_4	Tridymite	421.83 ($Z = 4$)	105.458
$\text{BaFe}_{12}\text{O}_{19}$	Hexagonal ferrite	696.20	348.10
$\text{PbFe}_{12}\text{O}_{19}$	Hexagonal ferrite	694.90	347.45
$\text{BaBiNb}_5\text{O}_{15}$	Tungsten-bronze	616.31 ($Z = 2$)	308.155

PbBiNb ₅ O ₁₅	Tungsten-bronze	2420.08 (Z = 8)	302.51
BaNb ₂ O ₆	Tungsten-bronze	248.91 (Z = 2)	124.455
PbNb ₂ O ₆	Tungsten-bronze	4905.25 (Z = 40)	122.631
BaTiO ₃	Perovskite	64.427	64.427
PbTiO ₃	Perovskite	63.229	63.229
Ba ₃ P ₂ O ₈	Palmierite	571.29 (Z = 3)	190.43
Pb ₃ P ₂ O ₈	Palmierite	692.78 (Z = 4)	174.195

I have the following comments on the manuscript.

While there is a large difference in the scattering factor of Ba and Pb, why the occupancy factor of Ba and Pb were fixed? The authors suggest the disorder is responsible for high thermal parameters for Ba/Pb site. But I see that the U is high for x =1 as well. Is it because of large asymmetric distortion around Pb-ions?

Response: It is a good question.

We did attempt to refine the occupancy factors and sites for Ba and Pb separately, however, such models did not give reasonable crystallographic parameters (bond length, particularly), which is very likely ascribe to Ba²⁺/Pb²⁺ local disorder for the compositions CaBa_{1-x}Pb_xZn₂Ga₂O₇ (0 < x < 1). The same disorder results in high correlation between ADPs and Ba/Pb occupancies. In the revised text we argue that since all the samples were found phase pure and metal disorder between (Ba,Pb) and (Ga,Zn) sites is implausible due to very large difference in size (>220%), the occupancy factors for Ba²⁺ and Pb²⁺ can be confidently fixed to the nominal values.

Reviewer #3 (Remarks to the Author):

The manuscript Unprecedented Lattice Volume Expansion on Doping Stereochemically Active Pb^{2+} into Rigid Structure of $\text{CaBa}_{1-x}\text{Pb}_x\text{Zn}_2\text{Ga}_2\text{O}_7$ reports extensive structural characterization of the title solid solution using X-ray and neutron diffraction/PDF supported by DFT calculations. The cell volume expansion claimed by the authors is certainly a peculiar behaviour of this original solid solution. The evidences laid out to support this surprising feature are convincing and the role of the Pb^{2+} lone pair is clearly identified by the authors. I believe that these results are of interest for the modification of all structures build upon BaO_3 and O_4 closed packing and therefore of great interest to the solid state community. However, in the present form I cannot recommend this article for publication in nature communication without major revision.

The structure of the present paper is a bit confusing, the problem in my opinion is that the three structural studies are presented separately. This results in three different structural models to describe the same structure. In the first part of the manuscript all the compositions are analyzed using XRD considering Ba/Pb and Ga/Zn mixed sites. Then in the second part of the manuscript, using neutron diffraction the authors are able to show that Ba and Pb are not on the same site (but keep correlated occupancies) and that the ratio of Zn/Ga is constant for the compositions $x = 0, 0.5, 1$. Finally, the orthorhombic distortion is used to explain the local structure by nPDF, but the authors do not mention which model fits best the long range data (according to the previous assumptions, the Pna21 model should be worse than $P6_3mc$). Therefore I suggest that all structural data be homogenized before publication to fit the most accurate model determined by XRD+NPD combined refinement or the nPDF eventually. Additionally, I would strongly recommend to used O3 as the origin of the polar space group rather than Ba/Pb to match with the text, i.e. Pb pops out of the plan instead of the whole layer moves away from Pb.

Response: First, thanks very much for your good suggestions. In the original manuscript, we tried to describe the unique phenomenon of lattice volume expansion and then to explain the origin step-by-step. However, we agree with the comment of Reviewer #3 and completely re-arranged the logic of the manuscript. We present 4 parts in “Results”, including (1) Unexpected cell volume expansion for $\text{CaBa}_{1-x}\text{Pb}_x\text{Zn}_2\text{Ga}_2\text{O}_7$ solid solutions, (2) Reciprocal space structure refinements, (3) Real space structure refinements, (3) Structure evolution. And in the final, a discussion is given.

Combined reciprocal and real space Rietveld refinements against TOF ND data and neutron PDF were further performed to verify the symmetry of the long-range structure. We tested both the $P6_3mc$ and $P31c$ models and found that the long-range structures for all compositions are better described using the $P6_3mc$ model rather than $P31c$ (see Figures S9-11). This result is consistent with the reciprocal refinements against both XRD and ND data.

Finally, following the suggestion of Reviewer, Rietveld refinements for all compositions were re-done by using O3 as the starting point of the polar space group $P6_3mc$. All the relevant figures and crystallographic data were updated. Given both the light and heavy atoms can be located precisely by combined refinement against XRD and ND data, we thus focused on the structure obtained the combined refinements.

The crystal chemistry of “114” oxides is not sufficiently described, the lower symmetry space groups ($P31c$ and $Pna2_1$) used by the authors have all been described for the cobaltite series $\text{LnBaCo}_4\text{O}_7$ with $\text{Ln} = \text{Ca}, \text{Y}, \text{Tb} - \text{Lu}$. In addition, $\text{LnBaFe}_4\text{O}_7$ with $\text{Ln} = \text{Y}, \text{Dy} - \text{Lu}$, a cubic polymorph, based on a different stacking of the same elemental layer, has also been described to go from $F4-3m$ to $I-4$ to $P2_1$. The numerous phase transitions observed in these systems, as well as the

local distortion evidence by nPDF, contradict the claim by the author of a “rigid structure” in the ab plan.

Response: It is really a good comment. “114” oxides such as $LnBaCo_4O_7$ and $LnBaFe_4O_7$ indeed exhibit Ln -size dependent structure symmetries. We revised the text focusing on the role of the uniaxial strain as the main driving force for the cell volume expansion of $CaBa_{1-x}Pb_xZn_2Ga_2O_7$ without breaking average symmetry.

The framework of $CaBaZn_2Ga_2O_7$ is highly strained along the c -axis, which can be deduced from the small c/a value (~ 1.605 vs. ideal 1.633) and the high over-bonding of Ca^{2+} (BVS ~ 2.48 vs. ideal 2). The incorporation of lone pair active Pb^{2+} into the lattice leads to a significant expansion of the c -axis and an overall cell volume expansion, which in fact is a macroscopic behavior of the release of the structure strain. More details of explanation are indicated in red in the revised text.

Few additional question and remarks

Fig 1: How is the size of the A-site calculated? And why is it so linearly dependent on the composition despite the accidents in the evolution of the cell parameters?

Response: The average A-site size is calculated in $CaBa_{1-x}Pb_xZn_2Ga_2O_7$ as $r(A^{2+}) = (1-x) r(Ba^{2+}) + x r(Pb^{2+})$. From this equation, we can deduce that the A-site size should be linearly dependent on x .

The calculation of A-site size was added to the caption of Figure 1.

Fig 2: The legends are not properly described. Both curves in Fig 2-a have the same legend, it might be necessary to differentiate the two O1 position. I suppose that is the average distance, but this is

not clearly stated. Finally, these data correspond to a structural model proved to be wrong by neutron powder diffraction, Ba and Pb do not occupy the same site...

Response: We agree and revised the figures to make them clearer. First, to differentiate the two O1 positions clearly, we labelled them with O1a and O1b, respectively. The surrounding oxygens atoms are also labelled clearly for [Ba/PbO₁₂] dodecahedron in Figure 5.

We also re-checked the original version of the combined refinements against ND and XRD data of $\text{CaBa}_{0.5}\text{Pb}_{0.5}\text{Zn}_2\text{Ga}_2\text{O}_7$ and found that Pb^{2+} located at $z \sim 0.45$, O3 located at ~ 0.45 , and Ba^{2+} located at $z \sim 0.5$. It was incorrect because this means Ba^{2+} displaced from the center of [Ba/PbO₁₂] dodecahedron, not Pb^{2+} . Thanks to the comments from reviewers, in the revised version, we corrected this point. Now the structure model with $\text{Ba}^{2+}/\text{Pb}^{2+}$ cations sharing the same atomic coordinate and anisotropic atomic displacements (ADPs) are applied to account for the $\text{Ba}^{2+}/\text{Pb}^{2+}$ disordering in $\text{CaBa}_{1-x}\text{Pb}_x\text{Zn}_2\text{Ga}_2\text{O}_7$ with $0 < x < 1$. This model gives stable Rietveld refinement convergences as well as reasonable crystal structures for all Pb^{2+} -containing compositions. Moreover, using this model is easy for us to evaluate the average displacements for $\text{Ba}^{2+}/\text{Pb}^{2+}$ (or the LP activity).

Fig 6: what is “the stereochemical lone pair activity”? I cannot find any details on this parameter.

Response: The stereochemical lone pair activity can be reflected by the displacement distance of $\text{Ba}^{2+}/\text{Pb}^{2+}$ cations. This is indicated in the figure caption.

Fig S8: I am not sure how this figure was obtained but if they are mode decomposition then the possible movement in $Pna2_1$ are a combination of the $P31c$ and $Pna2_1$ space group.

Response: In the original manuscript, Fig. S8 was used to describe the differences of the $P6_3mc$, $P31c$ and $Pna2_1$ models. In the $P31c$ and $Pna2_1$ models, oxygen atoms (O3) in the $[BaO_3]$ layers are not confined by the mirror plane, which thus allow the oxygen atoms displace freely within the ab -plane. The black arrows in Fig. S8 showed only the displacement **directions** of oxygen atoms, which are observed for “114” oxides $SrBaZn_2Ga_2O_7$ ($P31c$) and $Sr_2Zn_2Ga_2O_7$ ($Pna2_1$) (*Inorg. Chem.*, **57**, 7770-7779 (2018)) rather distortion mode amplitudes. To avoid similar confusion of the readers, we omitted the figure in the revised manuscript and elaborated in the text instead.

Reviewers' comments:

Reviewer #1 (Remarks to the Author):

Jiang and co-workers have re-submitted their work on the unconventional volume expansion observed for the complex oxide solid solution $\text{CaBa}_{1-x}\text{Pb}_x\text{Zn}_2\text{Ga}_2\text{O}_7$. In line with the previous recommendation, the findings reported by Jiang et al do merit publication in Nature Communication. The authors have considered all the comments from the three reviewers in a constructive manner, and particularly the structure and clarity have been considerably improved. In conclusion, the paper is recommended for publication in Nature communication in the revised form.

Reviewer #3 (Remarks to the Author):

The revised version of the article "Unprecedented Lattice Volume Expansion on Doping Stereochemically Active Pb^{2+} into Uniaxially Strained Structure of $\text{CaBa}_{1-x}\text{Pb}_x\text{Zn}_2\text{Ga}_2\text{O}_7$ " by Jiang et al. still focuses on explaining the surprising contraction of the out of plane axis upon Pb^{2+} substitution. As mentioned by another referee while this is the kind of oddities one can expect due to the lone pair of Pb^{2+} vs. the spherical nature of Ba^{2+} , this behaviour is actually uncommon as clearly evidenced by the addition of table S2 in the revised manuscript. Just like in the original version of the article, the structural evidence presented in the article clearly supports the main claim of the article. Furthermore, the modifications of the revised version are beneficial to the general understanding of the paper.

However, I would like to point out that the authors decided to keep the XRD and NPD refinements as two separate studies. This results in the presentation of two (slightly) different structures for the same compositions ($x = 0, 0.5$ and 1 , see Fig. 6 for example). Similarly to my review of the original version, I still find it confusing to use the less accurate XRD results to build a model that is refined in a second step using more accurate XRD + NPD combined refinement. Please note that the small discrepancies between the two models will not influence the general conclusion. It simply results (in my opinion) in the presentation of twice as much structural data than required and the additional confusion coming from the fact that the authors do not clearly chose between the two models (reporting both in Fig. 6 for example). As the authors seem to disagree with me on this point, I leave the editor to decide whether this is an issue worth delaying the publication of this article. This point aside, I recommend the publication of this article in Nature Communication after the correction of the few minor errors listed below.

P3 "A long-range ordered alignment of such asymmetric units may break the inversion symmetry of the structure, leading to a polar structure with intriguing properties"
Losing an inversion centre does not necessarily results in a polar structure see P-43m (#215) for an example of acentric non-polar space group.

P5 "The A-site size values are calculated according to the equation: $r(\text{A}^{2+}) = (1-x) r(\text{Ba}^{2+}) + x r(\text{Pb}^{2+})$
This formulae estimates the average radius of a Ba/ Pb^{2+} cation or average radius of the A-site cation. The size of the A-site in itself should be calculated based on the cell parameter of the structure.

P5 " $\text{CaBa}_{1-x}\text{Pb}_x\text{Zn}_2\text{Ga}_2\text{O}_7$ crystallize in the so-called "114"-type oxide structure, where the [BaO3] and [O4] layers form a mixed cubic and hexagonal closed packing ionic structure, ..."
I assume the authors mean a mixture of ABC-ABC (cubic) and AB-AB (hexagonal) closed packing stacking? The actual sequence should be used here.

P7 "In the composition with the highest x , it is Pb^{2+} that controls the crystal structure geometry and c-axis becomes too long for Ba^{2+} and ADP becomes a flattened ellipsoid within the ab-plane, so

CaPbZn₂Ga₂O₇ was described with a site splitting model"

It is not clear why the author chose to deal with flattened ellipsoid using a splitting model while elongated ellipsoid are not accounted with the same kind of model.

P7 "All these observations in CaBa_{1-x}Pb_xZn₂Ga₂O₇ are thus different from other isostructural oxides without stereochemically active cations, e.g. MAZn₂Ga₂O₇ (M = Ca²⁺, Sr²⁺; A = Ba²⁺, Sr²⁺), where successive symmetry lowering and cation disorder-order transitions were observed.¹⁸"
The transition temperature might simply be lower than room temperature in both cases, does not mean it is not there. Actually, nPDF strongly suggest otherwise.

P10 "As shown in Figure S5a, some Bragg peaks of the normalized structure functions for CaBa_{1-x}Pb_xZn₂Ga₂O₇ (x = 0, 0.5 and 1) display a shift to lower Q as Pb-content increases."
Evolution of the cell parameter were already discussed in the first part of the article.

P10 "Apart from the change of peak positions, another noteworthy feature is the obvious decrease or increase in the magnitude of the atomic pairs with increasing Pb²⁺-content, indicating a wide range of atomic distributions"

Which pairs or distances are we talking about?

P10 "(Zn/Ga)₄" ⇒ (Zn/Ga)O₄

P11 "Such a local structure distortion unambiguously ascribes to Zn²⁺/Ga³⁺ disordering, rather than the Pb²⁺-to-Ba²⁺ substitutions"

I am not sure I understand this sentence correctly, the authors mean that the reason of the distortion of the local structure is the Zn/Ga disorder? Why would that be the case? Because the distortion is also evidence for x = 0?

P11 "It was recently reported that a similar cation inversion disordering induced a local structure distortion in spinel Mg_{1-x}NixAl₂O₄ and CuMn₂O₄,^{24,25}"

I know what cation inversion is in the spinel structure but to my knowledge that remain to be defined in the 114 structure. I do not think the author are claiming that Ca and Zn/Ga are migrating from Oh to Td and vice versa?

P11 "Here, the refined nanoscale structures for CaBa_{1-x}Pb_xZn₂Ga₂O₇ (x = 0 and 1) (see Figure S7) demonstrate that the local orthorhombic distortion is not significant."

Not significant on the long range I suppose?

Reviewers' comments:

Reviewer #1 (Remarks to the Author):

Jiang and co-workers have re-submitted their work on the unconventional volume expansion observed for the complex oxide solid solution $\text{CaBa}_{1-x}\text{Pb}_x\text{Zn}_2\text{Ga}_2\text{O}_7$. In line with the previous recommendation, the findings reported by Jiang et al do merit publication in Nature Communication. The authors have considered all the comments from the three reviewers in a constructive manner, and particularly the structure and clarity have been considerably improved. In conclusion, the paper is recommended for publication in Nature communication in the revised form.

Response: Thanks to the recommendation.

Reviewer #3 (Remarks to the Author):

The revised version of the article “Unprecedented Lattice Volume Expansion on Doping Stereochemically Active Pb^{2+} into Uniaxially Strained Structure of $\text{CaBa}_{1-x}\text{Pb}_x\text{Zn}_2\text{Ga}_2\text{O}_7$ ” by Jiang et al. still focuses on explaining the surprising contraction of the out of plane axis upon Pb^{2+} substitution. As mentioned by another referee while this is the kind of oddities one can expect due to the lone pair of Pb^{2+} vs. the spherical nature of Ba^{2+} , this behaviour is actually uncommon as clearly evidenced by the addition of table S2 in the revised manuscript. Just like in the original version of the article, the structural evidence presented in the article clearly supports the main claim of the article. Furthermore, the modifications of the revised version are beneficial to the general understanding of the paper.

Response: Thanks very much for the positive comments.

However, I would like to point out that the authors decided to keep the XRD and NPD refinements as two separate studies. This results in the presentation of two (slightly) different structures for the same compositions ($x = 0, 0.5$ and 1 , see Fig. 6 for example). Similarly, to my review of the original version, I still find it confusing to use the less accurate XRD results to build a model that is refined in a second step using more accurate XRD + NPD combined refinement. Please note that the small discrepancies between the two models will not influence the general conclusion. It simply results (in my opinion) in the presentation of twice as much structural data than required and the additional confusion coming from the fact that the authors do not clearly chose between the two models (reporting both in Fig. 6 for example). As the authors seem to disagree with me on this point, I leave the editor to decide whether this is an issue worth delaying the publication of this article. This point aside, I recommend the publication of this article in Nature Communication after the correction of the few minor errors listed below.

Response: We appreciate the comment. Indeed, there are very small differences between the structural models from XRD and XRD+NPD data. The later one is obviously more accurate. To eliminate any confusion and unnecessary repeat, we decide to just use the accurate one. The captions of Figs. 5 and 6 give the additional explanations. For example, to show the evolution trend of the bond distances in Fig. 5a, the parameters for the samples with $x = 0, 0.5$ and 1 were obtain from the combined refinements (Cu $K\alpha_1$ XRD + NPD), while the parameters for the samples with other x values were obtained from the refinements on Cu $K\alpha$ XRD data. Hope it is clear now.

Fig. 5 Evolution of Ba/Pb–O bond lengths and Ba/Pb displacements. Plots of $\text{Pb}^{2+}/\text{Ba}^{2+}\text{--O1}$ bond lengths (a), $\text{Pb}^{2+}/\text{Ba}^{2+}\text{--O3}$ and average $\langle \text{Pb}^{2+}/\text{Ba}^{2+}\text{--O} \rangle$ bond lengths (b), and average $\text{Pb}^{2+}/\text{Ba}^{2+}$ displacement distance (c) against the Pb-content in $\text{CaBa}_{1-x}\text{Pb}_x\text{Zn}_2\text{Ga}_2\text{O}_7$. The inset shows the coordination environment of $\text{Ba}^{2+}/\text{Pb}^{2+}$ in $\text{CaBa}_{1-x}\text{Pb}_x\text{Zn}_2\text{Ga}_2\text{O}_7$ ($0 < x < 1$) obtained from Rietveld refinements against XRD data with the $P6_3mc$ model, where a highly anisotropic thermal motion along c -axis can be visually observed. For $x = 0, 0.5$ and 1 , the parameters are obtained from combined Rietveld refinement against both XRD and ND data.

Fig. 6 Plots of c/a values against (a) the doping Pb-content and (b) the stereochemical lone pair activity (average displacements for $\text{Ba}^{2+}/\text{Pb}^{2+}$ cations) in $\text{CaBa}_{1-x}\text{Pb}_x\text{Zn}_2\text{Ga}_2\text{O}_7$. For $x = 0, 0.5$ and 1 , the parameters are obtained from combined Rietveld refinement against both XRD and ND data.

P3 “A long-range ordered alignment of such asymmetric units may break the inversion symmetry of the structure, leading to a polar structure with intriguing properties” Losing an inversion centre does not necessarily results in a polar structure see $P-43m$ (#215) for an example of acentric non-polar space group.

Response: We have changed our representation in the main text and also given below.

“A long-range ordered alignment of such asymmetric units may break the inversion symmetry of the structure, possibly leading to a polar structure with intriguing properties”

P5 “The A-site size values are calculated according to the equation: $r(A^{2+}) = (1-x) r(Ba^{2+}) + x r(Pb^{2+})$

This formula estimates the average radius of a Ba/Pb²⁺ cation or average radius of the A-site cation.

The size of the A-site in itself should be calculated based on the cell parameter of the structure.

Response: Thanks for pointing out this mistake, we revised the “A-site size” into “average radius of A-site cation”.

Fig. 1 Lattice parameters. Plots of the average A-site size and refined lattice parameters for $CaBa_{1-x}Pb_xZn_2Ga_2O_7$ along with the increasing Pb-content. The average radius of A-site cation is calculated according to the equation: $r(A^{2+}) = (1-x) r(Ba^{2+}) + x r(Pb^{2+})$.

P5 “ $CaBa_{1-x}Pb_xZn_2Ga_2O_7$ crystallize in the so-called “114”-type oxide structure, where the $[BaO_3]$

and [O₄] layers form a mixed cubic and hexagonal closed packing ionic structure, ...” I assume the authors mean a mixture of ABC-ABC (cubic) and AB-AB (hexagonal) closed packing stacking? The actual sequence should be used here.

Response: Revised accordingly. The changes are indicated in red in the main text and also below.

“CaBa_{1-x}Pb_xZn₂Ga₂O₇ crystallize in the so-called “114”-type oxide structure, where the [BaO₃] and [O₄] layers form a mixed cubic and hexagonal closed packing ionic structure **with the stacking sequence of -CBABC-**, leaving the octahedral and tetrahedral cavities partially occupied by Ca²⁺ and Zn²⁺/Ga³⁺ cations, respectively.¹⁸”

P7 “In the composition with the highest x, it is Pb²⁺ that controls the crystal structure geometry and c-axis becomes too long for Ba²⁺ and ADP becomes a flattened ellipsoid within the ab-plane, so CaPbZn₂Ga₂O₇ was described with a site splitting model” It is not clear why the author chose to deal with flattened ellipsoid using a splitting model while elongated ellipsoid are not accounted with the same kind of model.

Response: First, the elongated ellipsoid in CaBa_{1-x}Pb_xZn₂Ga₂O₇ for x = 0.2-0.9 were used to account for the Ba²⁺/Pb²⁺ disordering (Pb²⁺ deviates from the center of [O₁₂] dodecahedron, while Ba²⁺ remain at the center). Thus, the observed elongated ellipsoid for Ba²⁺/Pb²⁺ in CaBa_{1-x}Pb_xZn₂Ga₂O₇ arises from the fact that Ba²⁺ and Pb²⁺ located at different sites in the microstructure with different occupancies, namely, 1-x and x. In the first version of our manuscript, we did attempt to deal with the Pb²⁺/Ba²⁺ disordering for x = 0.2-0.9 using a site splitting model with Ba²⁺ and Pb²⁺ locate at different sites. However, Rietveld refinements led to inappropriate structure models with Ba²⁺ deviate from the center of [O₁₂] dodecahedron rather than Pb²⁺. Finally, we thus used the anisotropic atomic displacements (ADPs) to account for the Pb²⁺/Ba²⁺ disordering.

Second, for $\text{CaPbZn}_2\text{Ga}_2\text{O}_7$, a flattened ADP was observed for Pb^{2+} , indicating Pb^{2+} deviates away from the 3-fold axis. A site splitting model with Pb^{2+} located at three symmetrical sites with an occupancy factor of 1/3 was thus used to account for the Pb^{2+} -disordering in $\text{CaPbZn}_2\text{Ga}_2\text{O}_7$. This site splitting model for $\text{CaPbZn}_2\text{Ga}_2\text{O}_7$ is consistent with the theoretical calculations that Pb^{2+} towards to one of O1 atoms to form covalency bond.

A simple explanation for using a site splitting model to account for the Pb^{2+} disordering in $\text{CaPbZn}_2\text{Ga}_2\text{O}_7$ were given in Supporting Information and also given below.

“For $\text{CaPbZn}_2\text{Ga}_2\text{O}_7$, a flattened ADP was observed for Pb^{2+} , indicating Pb^{2+} deviates from the 3-fold symmetry. Considering that Pb^{2+} deviates from the center of $[\text{PbO}_{12}]$ dodecahedron to form strong covalency bond with O1, which is revealed by our theoretical calculations, a splitting model with Pb^{2+} towards to one of the O1 atoms was thus used to account for Pb^{2+} disordering in $\text{CaPbZn}_2\text{Ga}_2\text{O}_7$.”

P7 “All these observations in $\text{CaBa}_{1-x}\text{Pb}_x\text{Zn}_2\text{Ga}_2\text{O}_7$ are thus different from other isostructural oxides without stereochemically active cations, e.g. $\text{MAZn}_2\text{Ga}_2\text{O}_7$ ($M = \text{Ca}^{2+}, \text{Sr}^{2+}$; $A = \text{Ba}^{2+}, \text{Sr}^{2+}$), where successive symmetry lowering and cation disorder-order transitions were observed.¹⁸” The transition temperature might simply be lower than room temperature in both cases, does not mean it is not there. Actually, nPDF strongly suggest otherwise.

Response: It is quite a good comment. We agree with the reviewer that $\text{CaBa}_{1-x}\text{Pb}_x\text{Zn}_2\text{Ga}_2\text{O}_7$ might show structural transition at low temperature. At this point, our representation is not correct. So, we revised our text, and I think the description in the revised manuscript is OK now.

“All these observations in $\text{CaBa}_{1-x}\text{Pb}_x\text{Zn}_2\text{Ga}_2\text{O}_7$ are thus different from other isostructural oxides without stereochemically active cations, e.g. $\text{MAZn}_2\text{Ga}_2\text{O}_7$ ($M = \text{Ca}^{2+}, \text{Sr}^{2+}$; $A = \text{Ba}^{2+}, \text{Sr}^{2+}$),

where successive symmetry lowering and cation disorder-order transitions were observed.¹⁸ Such a symmetry and cation disorder-order transitions might be observed for $\text{CaBa}_{1-x}\text{Pb}_x\text{Zn}_2\text{Ga}_2\text{O}_7$ at lower temperatures.”

P10 “As shown in Figure S5a, some Bragg peaks of the normalized structure functions for $\text{CaBa}_{1-x}\text{Pb}_x\text{Zn}_2\text{Ga}_2\text{O}_7$ ($x = 0, 0.5$ and 1) display a shift to lower Q as Pb-content increases.”

Evolution of the cell parameter were already discussed in the first part of the article.

Response: Thanks for your suggestion. We deleted this repeated presentation. The changes are indicated in red in the main text and also given below.

“The normalized structure functions and nPDFs for $\text{CaBa}_{1-x}\text{Pb}_x\text{Zn}_2\text{Ga}_2\text{O}_7$ ($x = 0, 0.5$ and 1) are presented in Figure S5. Apparent shift of the positions of Pb/Ba–O and Zn/Ga–Zn/Ga pairs are observed for the nPDFs (see Figure S5b), which is in line with the results deduced from XRD and ND.”

P10 “Apart from the change of peak positions, another noteworthy feature is the obvious decrease or increase in the magnitude of the atomic pairs with increasing Pb^{2+} -content, indicating a wide range of atomic distributions” Which pairs or distances are we talking about?

Response: Thanks for your suggestion. The related atomic pairs are indicated clearer in the revised manuscript and also given below.

“Apart from the change of peak positions, another noteworthy feature is the obvious decrease or increase in the magnitude of the atomic pairs with increasing Pb^{2+} -content, i.e. Zn/Ga–O and Ca–O pairs, indicating a wide range of atomic distributions. This observation suggests the Pb^{2+} -doping may lead to a local structure distortion.”

P10 “(Zn/GaO)₄” ⇒ (Zn/Ga)O₄

Response: Revised accordingly.

P11 “Such a local structure distortion unambiguously ascribes to Zn²⁺/Ga³⁺ disordering, rather than the Pb²⁺-to-Ba²⁺ substitutions” I am not sure I understand this sentence correctly, the authors mean that the reason of the distortion of the local structure is the Zn/Ga disorder? Why would that be the case? Because the distortion is also evidence for $x = 0$?

Response: The reciprocal refinements based on nPDFs revealed both Pb-containing and Pb-free compounds possess locally distorted structure, we thus can deduce that the local structure distortion is not induced by Pb²⁺-doping. This reason is given in the revised manuscript and also below.

“Because both Pb²⁺-containing and -free compounds are locally distorted, such a local structure distortion unambiguously ascribes to Zn²⁺/Ga³⁺ disordering, rather than the Pb²⁺-to-Ba²⁺ substitutions.”

P11 “It was recently reported that a similar cation inversion disordering induced a local structure distortion in spinel Mg_{1-x}Ni_xAl₂O₄ and CuMn₂O₄,^{24,25}” I know what cation inversion is in the spinel structure but to my knowledge that remain to be defined in the 114 structure. I do not think the author are claiming that Ca and Zn/Ga are migrating from O_h to T_d and vice versa?

Response: We do not mean that the local structural distortion stems from the anti-site disordering between Ca and Zn/Ga. The anti-site disordering between Ca and Zn/Ga is impossible because of their large difference in cationic size and occupancy preference.

Herein CaBa_{1-x}Pb_xZn₂Ga₂O₇, the T2-site cations are located at 6c sites, which is

symmetry-related by the 3-fold axis symmetry. Owing the Zn/Ga disordering in T2 site, the 3-fold axis symmetry is broken in the local structure. Moreover, the O1 atoms (and O3 atoms) are also related by the 3-fold axis symmetry. The difference in bond length between Zn-O and Ga-O bonds will also drives O1 atoms (and O3 atoms) deviates from the 3-fold axis symmetry. These local structure distortions could be readily detected by nPDFs. In the $Pna2_1$ model, there are 7 crystallographic independent oxygen atoms and 4 crystallographic independent T-sites to account for Zn²⁺/Ga³⁺ disordering induced local structure symmetry lowering. We can conclude that local structure distortion ascribes to the break of the 3-fold axis symmetry induced by Zn²⁺/Ga³⁺ disordering. So, it is understandable that both $P6_3mc$ and $P31c$ are not adequate to describe the local structure of CaBa_{1-x}Pb_xZn₂Ga₂O₇.

A paragraph was added in the Supporting Information to explain the observed local structure distortion in CaBa_{1-x}Pb_xZn₂Ga₂O₇, and also given below.

“In the $P6_3mc$ and $P31c$ models, O1, O3, and T2 atoms are located at the 6c sites. In the microscopic structure, the 3-fold axis should be not retained due to the Zn²⁺/Ga³⁺ disordering in T2 sites. Additionally, owing to the large difference in cationic radius between Zn²⁺ and Ga³⁺, the differences in distance between Zn-O and Ga-O pairs also drives O1 atoms (and O3 atoms) deviate from 3-fold axis symmetry. In other words, O1 atoms (O3 atoms, and T2 site atoms) are not strictly related by the 3-fold axis in the local structure because of the Zn²⁺/Ga³⁺ disordering and large difference in cationic size. In contrast, in the $Pna2_1$ model, there are 7 symmetry-independent oxygens and 4 symmetry-independent T-site atoms (Zn/Ga) to describe the local structural distortion, due to absence of 3-fold axis symmetry. Finally, we can conclude that the local structure distortion observed in CaBa_{1-x}Pb_xZn₂Ga₂O₇ is not induced by Pb-doping but Zn²⁺/Ga³⁺ disordering. As indicated by our real space refinements, such Zn²⁺/Ga³⁺ disordering induced local structural

distortions could be readily detected by nPDFs.”

P11 “Here, the refined nanoscale structures for $\text{CaBa}_{1-x}\text{Pb}_x\text{Zn}_2\text{Ga}_2\text{O}_7$ ($x = 0$ and 1) (see Figure S7) demonstrate that the local orthorhombic distortion is not significant.” Not significant on the long range I suppose?

Response: It is true that the distortion on the long range is also insignificant. This is why the combined real space and reciprocal Rietveld refinements manifested that both the long-range structure for $\text{CaBa}_{1-x}\text{Pb}_x\text{Zn}_2\text{Ga}_2\text{O}_7$ can be described with the high symmetry space group $P6_3mc$. A sentence about this add in main text and also given below.

“Here, the refined nanoscale structures for $\text{CaBa}_{1-x}\text{Pb}_x\text{Zn}_2\text{Ga}_2\text{O}_7$ ($x = 0$ and 1) (see Figure S7) demonstrate that the local orthorhombic distortion is not significant, further indicating the distortion on the long range would be also insignificant.”

REVIEWERS' COMMENTS:

Reviewer #3 (Remarks to the Author):

The questions raised in the precedent round of review have been satisfactorily answered by the authors and you can go ahead with the publication of the article "Unprecedented Lattice Volume Expansion on Doping Stereochemically Active Pb²⁺ into Uniaxially Strained Structure of CaBa_{1-x}Pb_xZn₂Ga₂O₇" by Prof Yang et al.

REVIEWERS' COMMENTS:

Reviewer #3 (Remarks to the Author):

The questions raised in the precedent round of review have been satisfactorily answered by the authors and you can go ahead with the publication of the article "Unprecedented Lattice Volume Expansion on Doping Stereochemically Active Pb^{2+} into Uniaxially Strained Structure of $\text{CaBa}_{1-x}\text{Pb}_x\text{Zn}_2\text{Ga}_2\text{O}_7$ " by Prof Yang et al.

Response: Thanks for your recommendation.